# Optic Neuritis in Multiple Sclerosis—A Review of Molecular Mechanisms Involved in the Degenerative Process

**Manuela Andreea Ciapă** [1], **Delia Lidia Șalaru** [2,3,*], **Cristian Stătescu** [2,3] , **Radu Andy Sascău** [2,3] and **Camelia Margareta Bogdănici** [4,5] 

1. Emergency Hospital Dimitrie Castroian, 735100 Huși, Romania
2. Cardiology Clinic, Institute of Cardiovascular Diseases, 700503 Iași, Romania
3. Department of Internal Medicine, University of Medicine and Pharmacy "Grigore T. Popa", 700115 Iași, Romania
4. Department of Surgical Specialties (II), University of Medicine and Pharmacy "Grigore T. Popa", 700115 Iași, Romania
5. Ophthalmology Clinic, Saint Spiridon Hospital, Iași 700111, Romania
* Correspondence: deliasalaru@gmail.com

**Abstract:** Multiple sclerosis is a central nervous system inflammatory demyelinating disease with a wide range of clinical symptoms, ocular involvement being frequently marked by the presence of optic neuritis (ON). The emergence and progression of ON in multiple sclerosis is based on various pathophysiological mechanisms, disease progression being secondary to inflammation, demyelination, or axonal degeneration. Early identification of changes associated with axonal degeneration or further investigation of the molecular processes underlying remyelination are current concerns of researchers in the field in view of the associated therapeutic potential. This article aims to review and summarize the scientific literature related to the main molecular mechanisms involved in defining ON as well as to analyze existing data in the literature on remyelination strategies in ON and their impact on long-term prognosis.

**Keywords:** inflammation; demyelination; remyelination; axonal degeneration; molecular mechanisms

## 1. Background

Multiple sclerosis (MS) is a demyelinating and neurodegenerative disease of the central nervous system (CNS), influenced by both genetic, (auto)immune, and environmental factors [1,2]. Paresthesia, motor deficit, autonomic spinal cord symptoms, visual symptoms, ataxia, exhaustion, disorientation, lack of sleep, discomfort, and depression are among the most prevalent symptoms. Structural and functional abnormalities in the visual system are targeted in most patients with MS, typically at the earliest stages of the disease, defining a hallmark feature of MS, namely optic neuritis (ON). The evolution of the clinical picture of patients with MS is extremely variable and heterogeneous in terms of locations and extensions of brain and spinal cord lesions [3]. ON is an inflammatory injury of the optic nerve that leads to visual disability. Unilateral visual acuity diminution, visual field loss, color vision deficiencies, diminished contrast, and brightness perception are frequent clinical manifestations of ON [4]. Recurrence of acute episodes of ON as well as chronic axonal injury causing structural changes over time are responsible for optic pathway damage [5].

ON is ubiquitous in the evolution of MS, up to 70% of patients with MS having an acute episode of ON during their course [6]. For 15–20% of patients with MS, the diagnosis of an acute episode of ON requires additional investigations that subsequently identify the underlying pathology [7]. In the first 6 months after diagnosis of MS, targeting an episode of ON induces a significant change in measurement of retinal nerve fiber layer (RNFL) thickness (a drop down to 20 μm) [8,9]. More and more studies in the field are addressed to

retinal measurements, identified in multiple researches as markers of neurodegeneration, with retinal damage already demonstrated 6 months after ON [10]. Calabia et al. [11] concluded on a similar clinical study that ON should not be regarded as a potential factor of clinical impairment in patients with MS.

Clinical studies have demonstrated that ON and MS associate the same characteristics of inflammatory demyelination, the substrate being perivascular infiltrates that induce a significant cellular response that secondarily causes myelin damage in the nerve parenchyma. On the other hand, ON and MS associate distinct pathophysiological mechanisms that support CNS immune involvement [3].

There is a growing interest in depicting intimate mechanisms of MS, starting with inflammation, demyelination, axonal degeneration and the possibility of remyelination, and the study of optic nerve pathology offers a promising perspective of understanding and, further on, extrapolating the physiopathological mechanisms in MS. Several clinical studies in the field give a leading role to inflammation and neurodegeneration in the development of central nervous system damage [12,13]. Injury to the optic nerve also causes optic neuropathy, an entity with neurodegenerative substrate that causes visual acuity impairment over time [14]. Several studies in the field have shown that neurodegeneration occurs early in patients with ON [15].

We conducted a search using PubMed and SCIENCE DIRECT in July 2022, using the terms and phrases, "optic neuritis", "multiple sclerosis", "inflammation", "molecular mechanisms", "axonal degeneration", "biomarkers" and "therapeutic targets" under different word associations. We focused on studies related to ON in MS (published between 1970 and July 2022), with an emphasis on future directions in research and treatment, and we explore the potential implications for improved management of disease progression.

## 2. Pathophysiology of Optic Neuritis, a Projection of MS Pathomechanism

In terms of MS pathophysiology, it is recognized that oligodendrocytes are responsible for myelination as well as for maintaining saltatory conduction to facilitate effective transmission of a nerve impulse down the axon in the CNS [5]. Damage to myelin (demyelination) and nerve fibers (axonal degeneration) in the CNS is the ultimate cause of MS. Immune cells are largely believed to assault myelinated axons in the CNS, resulting in demyelination and axonal degeneration [7]. Activated autoreactive T cells, myelin-specific T cells, B cells, plasma cells, dendritic cells, and macrophages, for example, can enhance macrophage recruitment by releasing different cytokines and chemokines [16]. Within the CNS, infiltrating inflammatory cells activate and interact with other immune cells and neuronal cells, resulting in oligodendroglial cell death-mediated demyelination, glial cell activation (including microglia and astrocytes), and axonal degeneration [17,18].

The structure of the anterior visual pathway is complex in which the retinal ganglion cells play a central role by positioning the nuclei at the level of the ganglion cell layer. Axons of the RNFL that are unmyelinated enter the optic nerve. Its path is through the optic canal to the level of the optic chiasm where the separation of the nasal fibers takes place. The synapse of most of the fibers takes place at the level of the lateral geniculate nucleus [19]. The role of immune mechanisms in the development and progression of inflammatory lesions of the optic nerve resides in understanding the anatomy and associated physiological mechanisms. The lamina cribosa separates the retina from the scleral wall of the eye socket, and is defined as a fibrous plaque composed of a dense network of collagen fibers. The nerve fibers within the lamina cribosa are non-myelinated. The location of oligodendrocytes in the posterior compartment explains the inflammatory status of the optic nerve during ON, as retinal inflammation is not typical of this ocular disorder [20].

### 2.1. Inflammatory Phase

The main inflammatory cells that are activated in an early stage of the inflammatory process in the brain are microglia, macrophages, and peripheral T lymphocytes. Activated T cells mature and expand clonally before dividing into effector cells and migrating through

the bloodstream to breach the blood–brain barrier (BBB). Endothelial cells in the CNS microvasculature contain adhesion molecules, which activated T cells can attach to and penetrate [3,18]. The release of cytokines and other proinflammatory mediators aggravates the inflammatory environment, attracting more immune cells to the CNS and eventually leading to demyelination [3,21,22]. Using a range of experimental animal models, the immunological processes underlying demyelination of the optic nerve secondary to the inflammatory process may be easily investigated [4]. The pathophysiological processes mentioned above are mediated by a variety of molecules with intrinsic action that potentiate the associated pro-inflammatory status [23].

The pathophysiological mechanisms involved in MS are based on the "inside-out" and "outside-in" theories, which have been intensively studied in the literature [24]. The first entity is based on the existence of a subsidiary primary degenerative process that determines in a secondary plan the activation of autoimmune mechanisms [25]. The "outside-in theory" of MS has been proposed, as opposed to the "inside-out hypothesis", according to which there is an autoimmune substrate that allows CD4+ T lymphocytes attack against myelin [25]. Antigen-presenting cells (APCs) reawaken autoreactive effector CD4 T cells in the CNS and attract more T cells and macrophages to develop the inflammatory lesion [26,27] (Figure 1).

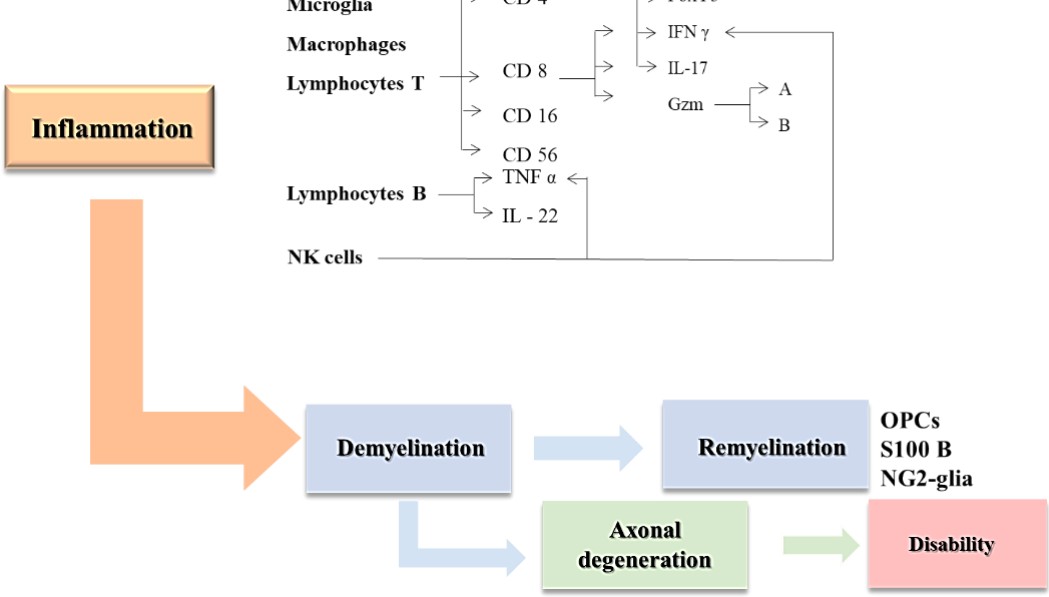

**Figure 1.** "Outside-in" theory (details in the text)-cells and processes involved.

CD4 T cells are identified in the cerebrospinal fluid (CSF) of MS patients and deep inside CNS lesions [26,28]. The activation of CD4 T lymphocytes is controlled by DR2 (DRB-1501/DQ6) is a major histocompatibility complex (MHC) class II locus [28]. Recent clinical studies in the field certified that Tr1 CD4+ regulatory populations are now recognized to control autoimmune T cell activity. Furthermore, reducing CD4 T cells would have little effect on CD8 T cells, which make up the bulk of CNS-resident T cells in patients and may play a critical role in the illness once CD4 T cells have started it [26]. CD4 T cells that secrete interferon gamma (IFNg) and interleukin-17 (IL-17) are thought to be the pathogenic initiators of MS.

CD4 T cells were originally divided into two functionally different types in the late 1980s: IFNg-producing Th1 cells that remove external infections and IL-4-producing Th2 cells that trigger allergic reactions [29,30]. Following that, researchers discovered CD4+ Th17 cells, which play a key role in autoimmunity. Tumor necrotic factor alpha (TNFα) promotes inflammation by activating STAT3 and IL-22 promotes inflammation by activating STAT3. They also have lower levels of IL-10, an anti-inflammatory cytokine [31].

CD8 T lymphocytes have an important role in MS in humans. CD8 T cells make up the bulk of T cells in the CNS perivascular infiltrate and at the periphery of CNS lesions in MS, unlike experimental autoimmune encephalomyelitis (EAE) [32]. The density of CD8+ T cells is 50 times higher than CD4+ T cells due to perivascular cells located at the periphery of active demyelinating plaques in patients with progressive MS. However, this ratio is not supported by CSF analysis where the ratio is at most 6:1 or by peripheral blood analysis where the ratio is much lower at only 2:1 [22]. CD8 T cells are also often seen in disease-related cortical plaques [33,34]. Some CD8+ T cell subtypes associate oligoclonal growth, thus being an indirect response of oligoclonal cell amplification to specific antigen responses [35].

CD8 T lymphocytes detect peptides of endogenous intracellular proteins given in the context of MHC class I molecules and destroy cells through a cell-contact-mediated mechanism involving granzyme A (GzmA) and granzyme B (GzmB) activities. While MHC class I is expressed ubiquitously and constitutively on all cells, MHC class I and class II expression is increased in oligodendrocytes, astrocytes, and neurons in patients with disease activity [36]. Antigen-presenting microglial cells have the ability to cross present foreign antigens on their MHC class I molecules, presumably resulting in the increased frequency of myelin-reactive CD8 T cells seen in MS patients [37].

IFNg is secreted by these myelin-reactive CD8 T cells, which destroy cells that express endogenously generated myelin. Because effector CD8 T cells' intracellular lytic granules are oriented toward adjacent axons in immunohistochemical investigation of postmortem CNS tissue slices, their cytotoxic action may play a key role in axonal injury. The presence of lesional CD8 T lymphocytes in close proximity to neurons has been linked to axonal damage [38]. In the setting of MS, at least two subgroups of CD4+ regulatory T (T-regs) cells have been discovered and examined. T-regs are a subset of regulatory T cells that express the transcription factor Forkhead box protein 3 (FoxP3) as well as a slew of inhibitory immune checkpoint surface molecules that help them suppress in vitro T cell proliferation via a cell-contact-mediated mechanism [39]. The Tr1 regulatory CD4+ T cell is a second kind of CD4+ regulatory T cell that controls cell proliferation predominantly through the release of IL-10 [40].

FoxP3+ Tregs, which account for fewer than 4% of circulating CD4 T cells, are known as "professional" suppressor cells because they prevent the activation of other cell types via a cell-to-cell contact mechanism [26]. Teff cells produced from patients is, in fact, immune to Treg-mediated repression [41]. According to published research, MS patients have both a deficiency in Tregs and a resistance to Treg suppression by Teff cells [26].

CD46, which substantially promotes IL-10, was used to stimulate CD4 T cells, and it was shown that MS CD4 T cells express less IL-10 than healthy CD4 T cells [26]. Because the expression of IL-10 and CD46 is increased in patients who react to IFNb treatment compared to cells from patients who do not respond, the ability of CD4 T cells to release IL-10 is related with lower disease activity in MS [42]. Il-10 secretion by non-pathogenic Th17 cells has also been observed to increase [43].

Natural killer (NK) cells release both pro-inflammatory (IFNg, TNFa) and anti-inflammatory (IL-4, IL-10) cytokines, and they' ve been linked to illness [26]. Although CD56 and CD16 cells, which account for 90% of NK cells in peripheral circulation, are cytotoxic right away, they are found in considerably lower numbers in tissue. CD56 bright cells, on the other hand, predominate in tissues, where they largely release cytokines and develop cytotoxic activity with time. Despite the fact that NK cells have been discovered in MS patients' demyelinating lesions, the majority of data suggests that NK cells play an immunoregulatory role in MS [32]. Immunomodulatory and immunosuppressive therapies increase CD56-bright NK cells, increases in NK frequency correlate with treatment response [44], decreased NK frequency has been linked to relapse [45], and in vitro NK functional activity increases during remission [26]. Untreated MS patients' CD56-bright NK cells show a lower capacity to limit the proliferation of autologous activated T cells, which might be related to CD56 bright NK cell malfunction as well as the discovery that MS CD4 T

cells are less susceptible to NK cell regulation [46]. This "NK resistance" by patient-derived T cells has been attributed to increased T cell production of the NK-inhibitory ligand HLA-E or lower CD155 expression on patient-derived T cells [47].

The presence of oligoclonal bands (OCBs) in the CSF, which are found in 95% of MS patients and are caused by clonally enlarged Ig-secreting cells, is a hallmark observation in the disease [48]. It was found that CSF OCB antibodies from four MS patients have specificity for a variety of ubiquitous intracellular proteins that are produced as debris during tissue breakdown. Although antibodies to myelin lipids in the CSF have been linked to severe MS, anti-lipid antibodies are also seen in systemic lupus erythematosus and Grave' s illnesses [26].

MS patients' brain parenchyma, meninges, and CSF contain clonally enlarged B lymphocytes, which are more common in the CNS early in the illness [49,50]. Increased B cell frequency in the CSF is linked to a faster course of the illness [51]. B cells in the CNS might have a role in MS by secreting chemokines/cytokines and presenting antigen to T cells, in addition to their possible capacity to make autoantibodies [26]. In lymph-node-like follicles located in the meninges, which are typically close to cortical lesions, B lymphocytes can cross the blood-brain barrier and become long-term CNS residents [52]. The presence of these follicle-like structures shows that B cells expand and differentiate into plasmablasts and plasma cells within the CNS itself [53].

Anti-CD20 also lowers the number of T cells in the blood and CSF by 20% and 50%, respectively [45], and the remaining T cells' ability to release IL-17 and IFNg [26]. Anti CD20 has a quick start of benefit because it eliminates a pro-inflammatory B cell fraction that induces T cell activation through antigen presentation or cytokine release [26].

### 2.2. MS Triad: Demyelination, Axonal Degeneration, and Remyelination

MS is defined by a progressive inflammatory status associated with demyelination and autoimmune neurodegeneration in the CNS. Numerous studies published in the literature demonstrate the concern of researchers in the field in understanding the pathophysiological mechanisms underlying the chronic evolution and irreversible disability [26]. Clinical studies published to date attribute the disabling outcome to persistent chronic inflammation, ongoing demyelination and failure to remyelinate, the latter two being major factors associated with a poor neurological prognosis. These 3 main pathophysiological pillars have been recognized as the building blocks of this pathophysiological triad in MS [54].

### 2.2.1. Demyelination and Axonal Loss

The persistence of a pro-inflammatory status causes axon loss over time secondary to demyelination. Axonal function is directly affected by the direct action of inflammatory cytokines, enzymes and nitric oxide, which are produced by activated immune cells. Remission of inflammatory processes may result in remyelination of surviving axons, although most EAE phenotypes are characterized by neuronal cell death due to associated inflammatory stress. Paraclinical targeting of the RNFL is a marker of irreversible axonal injury [3]. T-cells interfere with the action of antibodies in the CNS contributing to the supplementation of demyelination secondary to inflammatory processes in EAE. The same effect is obtained by injecting the antibody into the brain with human complement. [3].

### 2.2.2. Animal Models Used to Study Remyelination

Endogenous remyelination ensures nerve conduction and prevents neurodegeneration, being a complex process involving various pathophysiological processes and representing a promising therapeutic strategy for the future [55]. The central role has been assigned to oligodendrocyte progenitor cells (aOPCs), which thus acquires potential therapeutic value. These cells mediate a variety of pathophysiological processes, including activation, migration, proliferation, or differentiation [56]. It has been shown that only mature aOPCs-derived from neonatal OPCs contributes to the remyelination process. These cells have the ability to reconstitute themselves and not to be replaced by neural stem cells [57,58].

The literature includes several clinical studies on animal models based on which remyelination has been studied. These animal models allow histopathological identification of the presence of remyelination processes, as the myelin state formed is thinner and shorter compared to that at the time of myelination [59,60]. Franklin et al. [61] highlights a number of factors that prevent remyelination in MS patients, such as altered aOPCs, a lack of pro-regenerative factors, or an excess of inhibitory factors or errors in aOPCs-mediated pathophysiological processes.

Experimental Autoimmune Encephalomyelitis

Experimental autoimmune encephalomyelitis is known as the animal model of MS and is frequently used for research purposes to further investigate pathophysiological mechanisms [16]. It is a CNS autoimmune illness that is deliberately generated in susceptible species such as rats and primates by vaccination with CNS-specific antigens, peptides derived from these antigens, or CNS tissue homogenates. Transfer of encephalitogenic CD4+ T cells from draining lymph nodes of animals vaccinated for active EAE induction into syngeneic animals can also cause the illness. However, the validity of EAE as an MS model has been called into question [62]. Several molecular and cellular processes of MS pathogenesis have been revealed in EAE research [63–65].

Demyelinating Illness Caused by the Virus Theiler's Murine Encephalitis

Theiler's murine encephalitis virus-induced demyelinating disease (TMEV-IDD) is a model frequently used in clinical trials in patients with various demyelinating diseases, including MS [66–68]. TMEV is a positive-stranded RNA virus from the Picornaviridae family (genus Cardiovirus). Between days 5 and 10 after injection, the TO subgroup of TMEV causes acute encephalitis. TMEV-IDD is of special relevance because it represents a hypothetical situation in people in which a virus is the primary contributor for CNS inflammation and demyelination.

Importantly, the clinical manifestations of TMEV-IDD are comparable to those seen in individuals with progressive MS, including stiffness, incontinence, extremity weakness, and, finally, paralysis [68]. Intrathecal antibody production has been seen in this model, which is similar to the oligoclonal bands detected in the CSF of MS patients [69]. The general objection against this model is that a non-human virus was utilized to simulate a human disease. Surprisingly, it was recently found that a human-TMEV recombinant virus might produce Vilyuisk encephalitis, a kind of encephalomyelitis [68]. Additional viruses, such as murine hepatitis virus, canine distemper virus, coronaviruses, and several retroviruses, are also being utilized in experimental animals to induce MS-like demyelinating illness [69–72].

The Role of Cuprizone or Other Toxins

Demyelination in mice caused by the copper chelator cuprizone is a useful tool for MS research [63–65]. Cuprizone consumption by mice results in early oligodendrocytes (ODC) mortality, activation of microglia/macrophages, and subsequent reversible demyelination [73]. This model is beneficial for researching demyelination and remyelination, as well as their relationship to axonal loss [74]. It is extremely important for the progression of type III and type IV MS lesions, where alterations in ODC appear to represent the key events in disease pathogenesis. In addition to cuprizone, additional toxins, such as ethidium bromide and lysolecithin, are employed to induce demyelination in experimental mice [74].

The Role of Lysophospholipid Lysophosphatidylcholine (LPC, Lysolecithin)

It has been used for decades to produce demyelination in animal models of multiple sclerosis. A recent investigation of LPC damage and homeostasis processes discovered that LPC nonspecifically altered myelin lipids and swiftly caused cell membrane permeability; LPC injury in mice was phenocopied by other lipid disrupting agents. A subsequent increase in LPC five days following the injection into white matter implies that the brain possesses mechanisms to buffer LPC, and albumin buffering greatly reduced LPC damage

in culture [75]. LPC application was compared to agarose-gel loaded LPC (AL-LPC) in mouse optic nerve behind the globe via a small surgery in an attempt to research new processes of demyelination and to assess new medicines. Agarose loading was employed to extend the length of LPC exposure and thereby accomplish long-term demyelination.

Visual evoked potentials (VEPs) recordings revealed a large increase in the latency of the P1 wave and a decrease in the amplitude of the P1N1 wave at the lesion locations, as well as severe demyelination and axonal damage. The optimized model demonstrated that both AL-LPC and LPC groups had extended demyelination, axonal degeneration, and retinal ganglion cell (RGC) loss; however, these diseases were more widespread in the AL-LPC group [76]. Furthermore, the therapeutic potential of many medicines has piqued the interest of researchers, beginning with animal models of generated demyelinating diseases. Among the key factors known to limit CNS regeneration are myelin associated inhibitory factors such as NogoA [77], myelin associated glycoprotein (MAG) [78], and oligodendrocyte myelin glycoprotein (OMgp) [79]. These elements connect to a common receptor known as Nogo receptor 1 (NgR1) [80]. A wide range of cells express this receptor, including neurons, OPCs, astrocytes, microglia, macrophages, dendritic cells, and neural precursor cells. Although the physiological implications of Nogo-A/NgR interaction among glial cells are unclear, Nogo-A expressed on oligodendrocytes may interact with NgR produced by reactive astrocytes and microglia/macrophages in active demyelinating lesions of MS [81].

### 2.2.3. Axonal and Neuronal Degeneration

By secreting IFNg and IL-17, pathogenic CD8 T cells may also contribute to the disease [82]. In a BBB model using human cells and in mice models, these IFNg-, IL-17, and GzmB-producing effector CD8 T cells may also experience increased endothelial transmigration [83]. As a result, CD8 T cells may not only induce oligodendrocyte mortality and neuronal injury once within the CNS, but they may also amplify IFNg- and IL-17-mediated disease [26].

After demyelination, what happens to the axon? Axonal degeneration and morphological alterations of axonal organelles, such as axoplasmic reticulum (AR)-like structures, were observed to precede morphological abnormalities of myelin in EAE animals. It was discovered that morphological alterations in myelin, as well as morphological changes in axonal organelles, cause axonal degeneration. Although further research is needed, it appears to be a strong link between twisted axons and axonal degeneration [16].

In EAE and acute human MS lesions, axonal degeneration with localized axonal swellings and mitochondrial abnormalities are prominent. It has been suggested that intra-axonal mitochondrial disease in localized axonal degeneration might be the first ultrastructural indicator of damage, occurring before axon shape changes. Axonal degeneration has been linked to mitochondrial failure in several investigations of autopsied human MS brains and in vitro models. Axonal diseases were also seen in myelin-associated glycoprotein-2,3-cyclic nucleotide 3-phosphodiesterase-null animals, demonstrating that oligodendrocyte–axon interactions are necessary for structural and functional modulation between myelin and axons.

A growing body of data implies that axonal degeneration in MS and EAE is triggered by axonal AR and mitochondrial dysfunction, which is followed by an increase in axonal $Ca^{2+}$ levels produced by AR and mitochondria. Axoplasmic reticulum $Ca^{2+}$ release produced subsequent degeneration of spinal neurons [84]. Furthermore, it was discovered that in EAE spinal cords, the intensity of a mitochondrial fission-related protein, Drp1/Dlp1, rose, whereas the intensity of a mitochondrial fusion-related protein (MFN) dropped [16].

Reduced adenosine triphosphate synthesis in demyelinated upper motor neuron axon segments disrupts ion homeostasis, causes $Ca^{2+}$ mediated axonal degeneration, and contributes to MS patients' increasing neurological impairment [16]. Glutamate excitotoxicity is one neurodegenerative process thought to be implicated in MS pathogenesis [85]. Because RGCs have a high density of dendritic glutamate receptors, they are especially sensitive to

elevated glutamate levels in the retina [86]. Over-stimulation of ionotropic glutamate receptors is thought to lead to prolonged intracellular calcium increases capable of activating downstream pathways leading to cell death, and their over-stimulation is thought to lead to prolonged intracellular calcium increases capable of activating downstream pathways leading to cell death [87]. It was found that ultrastructural alterations in RGC axons of the optic nerve, as well as elongation of nodes of Ranvier, were observed at the outset of illness [85].

A loss in visual acuity and changes in the optic nerve cytoskeleton (as evidenced by modifications in actin treadmilling and expression of its regulatory proteins) occur during the induction phase of autoimmune optic neuritis (AON), as well as RGC degeneration, which may be replicated by intravitreal glutamate injection. Sühs et al. [88] demonstrated that intravenous administration of the NMDA receptor blocker MK-801 during the induction phase of AON causes activation of NMDA receptors before the onset of demyelinating optic nerve lesions associated with the inflammatory status associated. Another group of investigators emphasizes the beneficial role of the retinal calcium increase during the induction phase, which potentiates the aforementioned effect, contributing to the restoration of visual integrity, the resumption of optic nerve actin dynamics as well as RGCs neuroprotection [89]. This is backed up by the fact that the retinal calcium level rises during AON at the same time. This points to the NMDA receptor as the most likely possibility, which leads us back to the "inside-out" theory [85] (Figure 2).

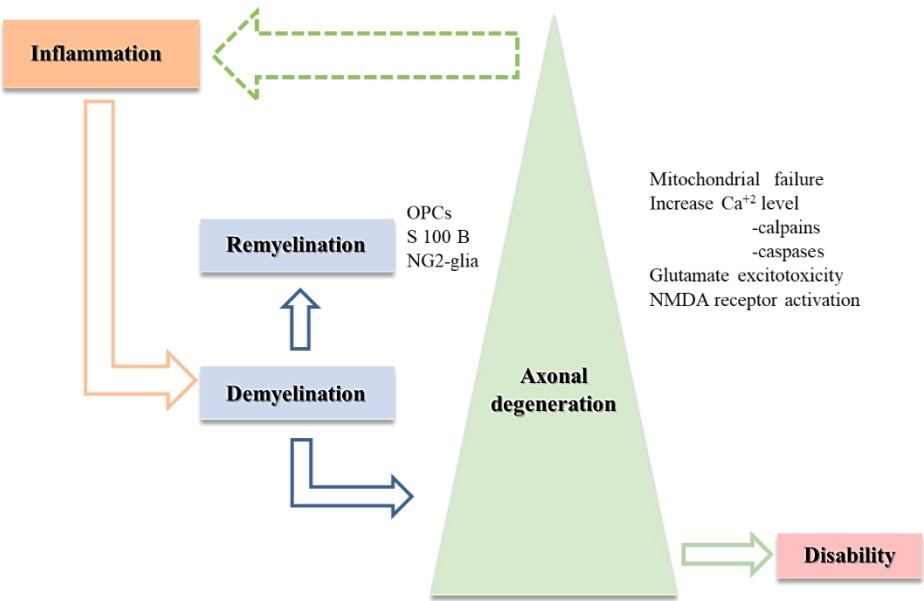

**Figure 2.** "Inside-out" theory (details in the text)–the green dotted arrow represents a possible temporal, but non-causal relationship.

Disturbances in the actin cytoskeletal dynamics of the optic nerve seen throughout the course of AON may have consequences for RGC degeneration since growing data indicates that actin is both a sensor and a mediator of apoptosis. F-actin disintegration is caused by both NMDA receptor activation [90] and increased intracellular calcium, resulting in actin network instability.

Calcium-dependent proteases, such as calpains and caspases, are activated by significant increases in intracellular calcium, further destabilizing the actin cytoskeleton. The actin-severing protein gelsolin is one such calcium-activated protease [91]. At the same time, calpain/caspase cleaves gelsolin [92], making the cell more sensitive to NMDA receptor activation as an anti-apoptotic agent [93]. A functional role in apoptotic signaling is also played by fractin, a calpain/caspase-cleaved actin monomer product that accumulates after activation of apoptosis [94]. Furthermore, the reorganization of nodes of Ranvier might be influenced by changes in actin network dynamics [85].

Gelsolin levels may be affected by NMDA receptor activation as well as other disease-related variables that are currently unknown. In contrast, while gelsolin protects cells against apoptosis [93] its expression might be up-regulated in response to glutamate-mediated stress, although for unknown reasons [85]. The early degeneration shown in this model, which occurs before the demyelination and inflammatory infiltration that define optic neuritis, contradicts the traditional belief that secondary RGC degeneration results from axonal injury in the demyelinated optic nerve [85].

Early AON retinal events might cause anterograde alterations in actin cytoskeletal dynamics in the optic nerve, which are most likely mediated by calcium build-up and activation of actin-regulatory proteases. NMDA receptor manipulation might be a therapeutically viable method for retinal neuroprotection in autoimmune neuro-inflammatory diseases.

### 2.2.4. Remyelination in Optic Neuritis

Immune-modulatory networks are activated, limiting inflammation, and initiating repair, resulting in at least partial remyelination and clinical remission [26]. S100B, a protein generated predominantly by astrocytes, has been shown to help with relapsing–remitting EAE. Administration of pentamidine isothionate (PTM) to EAE-induced mice abolishes S100B activity causing in a secondary plan improvement of preclinical scores, increase of remission rate and decrease of activity of some molecules present in the brain, such as IFNg, TNFa, or NOS activity. When comparing EAE animals treated with PTM to EAE mice not treated with medication, the number of CD68+ cells and demyelinating lesions were lower in PTM-treated EAE mice. Overall, this research implies that the severity of EAE is reduced by targeting neurotoxic mediators released by astrocytes [95].

MS pathophysiology is characterized by demyelination. NG2-glia are oligodendrocyte progenitors that can develop into adult oligodendrocytes and hence may help individuals with MS remyelinate [95]. The phenotypic heterogeneity of NG2-glia in relation to their ontogenic origin was investigated, as well as whether EAE causes a clonal NG2-glial response. They discovered that NG2-glia from single progenitors are distributed clonally across the grey and white matter [95].

The proliferative oligodendrocyte progenitor cell has been reported as the most effective remyelinating cell in animal experiments for successfully repairing demyelinating lesions, particularly those of the optic nerve [96]. The existence of a comparable population of oligodendrocyte progenitor cells in normal adult human white matter, as well as in acute and chronic MS lesions, may be the source of oligodendrocyte proliferation after demyelinating lesions in humans [97]. Although the presence of oligodendrocyte progenitor cells impacts the eventual number of oligodendrocytes in a demyelinating lesion, it does not appear that the quantity of oligodendrocytes is the sole component required for effective remyelination [98]. Some oligodendrocytes in acute MS lesions may reveal mild, early pathologic abnormalities, indicating that their myelinating capacity has been reduced without overt cell death [99]. Endogenous remyelination after ON appears to be most prominent in optic nerve lesions that develop early in the course of MS, and when significant remyelination occurs, it usually becomes morphologically apparent at least 1 month after the initial insult, a time interval that corresponds to clinical recovery after isolated typical ON [100]. Shadow plaques, which are made up of sparsely myelinated axons, are hypothesized to be the result of remyelination following a single bout of acute demyelination [98].

Recurrent demyelinating optic nerve damage in the same region of white matter, on the other hand, may impair those reparative processes, resulting in permanently demyelinated axons and failure of remyelination [100]. This discovery explains why remyelination is seen early in the course of MS but not in typical chronic MS lesions, which are more likely to have had several, temporally different bouts of demyelination [101]. While beneficial, endogenous remyelination in ON and MS in general has limits [98]. When compared to normal axons, remyelinated axons have thinner myelin sheaths and shorter internodal lengths [102]. Remyelinated axons, on the other hand, have poor axonal conduction

velocities [103]. Finally, lack of full remyelination is a major reason in the persistence of visual impairment after ON [98].

Jäkle et al. [104] performed an autopsy study on human brains from patients with MS and from unaffected controls, demonstrating both a reduced presence of ODC in shadowing lesions as well as changes in gene expression between areas of normal-appearing white matter of MS patients compared to a group of healthy subjects, raising the need for further clinical studies to understand the global cellular changes targeted in this category of patients. These results raise the observation that there are discrepancies between studies on animal models and those on humans, and that a comprehensive, potentially therapeutic approach is needed that addresses not only differentiation process [55,105,106].

The possibility of identifying therapeutic agents that offer neuroprotection to MS patients by potentiating remyelination has led to the emergence of various clinical trials—some ongoing, others completed—predominantly involving three types of agents: small molecules, hormones, and antibodies [107,108]. The most advanced clinical trials are hormone-based trials, most of which are Phase III clinical trials. Of all the incriminating agents, special attention is needed in the case of rHIgM22 (a remyelinating antibody) [109], which is currently the only agent acting on both OPCs and oligodendrocytes, leading to stimulation of acute and chronic myelination in preclinical models of demyelination [107,110].

The constant concern of researchers in the field to develop new molecules with remyelinating action has led to the emergence and conduct of multiple clinical trials whose interim results have already been presented in the literature. Thus, clemastine is an antihistamic agent acting on antimuscarinic receptors, with proven effects both in vitro and in vivo to date. It was tested in a Phase II randomized double-blind crossover placebo-controlled clinical trial (ReBUILD study), with the mechanism of action being the potentiation of OPC differentiation and proliferation. Preliminary results reported a shortening of P100 VEP latency by 1.7 ms/eye, indicating slightly faster neural transduction within the optic pathway but at the cost of fatigue as an adverse effect [111].

Olesosime is a cholesterol-like agent whose neuroprotective effect is exerted via mitochondrial metabolism. In vitro it induced maturation of OPCs and stimulation of myelin production. In the literature, there is a phase IB multicenter randomized double-blind placebo controlled clinical trial conducted to test the efficacy of this agent, but no superior results were observed compared to placebo [112–114]. In recent years, several phase II clinical trials have been conducted in which various therapeutic agents have been tested, such as bexarotene a retinoid x receptor γ (clinical trial in UK - EudraCT 2014–003145-99) [115,116], gold nanocrystals (stimulates ATP production by oxidizing nicotinamide adenine dinucleotide NADH to NAD+) [117,118], or domperidone (peripheral dopamine D2 receptor antagonist that stimulates prolactin secretion from the pituitary gland) [119,120] that stimulated remyelination by potentiating proliferation, differentiation, or maturation of OPCs, but they did not prove effective.

Opininumab is an anti-LINGO1 monoclonal antibody, which functions as a transmembrane protein at the OPCs and neuronal cell surface. Although the monoclonal antibody against LINGO1 has been shown to be effective in a phase I clinical trial, clinically significant results regarding visual acuity, VEP latency, or MRI measurements have not been demonstrated in several phase II clinical trials [121–124].

## 3. The "Big" Picture behind the MS Triad

Neuroimaging, CSF examination and VEP analysis are the main methods to establish the diagnosis of ON and assess the associated risk of developing MS.

### 3.1. MRI

ON is frequently the initial presentation of MS patients with no neurological history, especially demyelinating pathologies [125]. Neuroimaging is a central piece in the diagnostic and therapeutic puzzle. To date, clinical research in the field has not revealed the presence of molecules with a prognostic role for these patients, which has led to a shift

of attention toward imaging explorations, especially MRI [126]. Structural imaging parameters quantified by MRI cannot distinguish between demyelination and axonal lesions produced within the central nervous system [127]. MRI targeting T2-hyperintense and gadolinium-enhancing of multiple lesions of the brain or spinal cord are arguments in favor of the present MS [128,129]. Within the first 20 days of visual acuity decline, 95% of patients with MS-associated ON show T1 gadolinium enhancement [130]. Existing clinical studies in the literature refute the existence of a correlation between the extent and severity of lesions identified on MRI and the rate of vision recovery [131].

Swanton et al. [132] demonstrated that the presence of spinal lesions has a disabling predictive value for patients who develop MS over time (72% risk) compared to those without identified lesions, where the risk of progression was estimated at 25% [125,133,134]. In addition to absence of lesions on MRI assessment, male sex, lack of typical symptoms and optic swelling are factors associated with a low risk of ON progression to MS [134].

Over time, patients with ON may associate subclinical demyelinating lesions in which the usual paraclinical evaluation (CSF and VEP analysis) does not reveal pathological changes, the definitive being the MRI imaging exploration. Lebrun et al. [135] demonstrated that patients without MRI lesions have a clinical conversion rate of 33% to clinically isolated syndrome in 5 years. The investigators have highlighted as associated risk factors VEPs abnormalities, youth, and gadolinium enhancement on follow-up MRI. McDonald criteria are widely used in patients at risk of progression to MS, the main radiological changes quantified being dissemination in space or time [125,136]. In a similar clinical study, Tintore et al. [137] confirms the prognostic role of MRI scanning in assessing the occurrence of MS, compared to the Poser criteria, with the new standards associating superior sensitivity and specificity.

Frohman et al. [8] investigated the role of MRI versus optical coherence tomography (OCT) and laser polarimetry methods in the assessment of RNFL thickness vs. brain measures and concluded that measurement of RNFL thickness and radius of the optic nerve are preferred in clinical studies due to identification of more pronounced differences between patients with MS and controls.

### 3.2. Visual Evoked Potentials Analysis

VEPs is part of the diagnostic work-up of patients with ON, including asymptomatic forms, being an alternative to MRI imaging exploration [138]. Clinical studies show that 65% of patients show changes in VEPs, which are a clinical reflection of demyelination in the afferent visual pathways [139]. The most common findings are increased latencies and reduced amplitudes and abnormal waveforms [140].

Prolonged latency measurements suggest subclinical demyelinating damage, while reduced wave amplitudes are the paraclinical expression of axonal degeneration and loss in MS patients [141]. The parameters obtained by measuring VEPs have predictive value as well, being indirect markers, directly proportional to the severity of MS [142,143].

Recent clinical studies in the field focused on the multifocal visual evoked potentials and its role in ON and MS [144]. The evaluation of these potentials allows obtaining anatomical data on the localization of particular lesions, thus facilitating the deciphering of pathophysiological mechanisms focused on the triad demyelination, atrophy and remyelination [145]. De Santiago et al. [146] evaluated multifocal VEPs from 15 patients with radiologically isolated syndrome and concluded that measuring signal-to-noise ratio increases the risk of identifying patients with a high risk of developing MS over time.

Multifocal VEP have therapeutic value, their evaluation being used in various clinical trials with remyelination therapies as end-points. Klistorner et al. [145] demonstrated that Opicinumab (a human monoclonal antibody) vs. placebo in patients with ON decreases the risk of long-term visual impairment after remission of the acute episode, having a satisfying safety and tolerability profile [123,124,147]. Both VEPs and multifocal VEP have proven diagnostic value in the clinical studies presented above, with the latter having superior sensitivity (95%) and specificity (90%) [148].

Klistorner et al. [149] also demonstrated that amplitude of waves measured by VEP correlates positively with RFNL thickness after an acute episode of ON, with the most significant structural changes in RFNL being at the temporal level. Laron et al. [150] demonstrated that multifocal potentials analysis provides superior prognostic data compared to Humphrey visual field and OCT in MS patients.

### 3.3. Cerebrospinal Fluid Examination

Cerebrospinal fluid (CSF) analysis has both diagnostic and therapeutic value, due to biomarkers with predictive value for the development of MS in patients with acute ON [151–154]. Research presented in the literature in recent years attests to the concern of researchers in identifying molecules with both a diagnostic and prognostic role, on the basis of which the disease activity or therapeutic response in patients with MS can be assessed [155].

Olesen et al. conducted a prospective study on 40 patients with ON of which 16 were diagnosed with MS during the 2.5-year follow-up period. The CSF analysis demonstrated that TNF-$\alpha$, IL-10, CXCL13, and NF-L correlates positively with the diagnosis of MS, thus raising the hypothesis of the existence of inflammatory and neurodegenerative processes that started earlier [139]. Based on the potential biomarkers identified, the same investigators proposed two models to predict ON patients' risk of developing MS. Statistical analysis of the proposed models revealed an associated risk of up to 10% of developing MS after an ON episode and up to 15% for potential biomarkers.

IL-10 is a cytokine with an anti-inflammatory and immunosuppressive role that mediates a variety of pathophysiological processes in various inflammatory pathologies, not only MS [156]. Previous studies concluded that IL-10 correlates with higher IgG levels in patients with positive oligoclonal IgG bands [157]. IL-10 also interferes with pathophysiological mechanisms involved in MS, mediated by B cells [158]. The presence of a pleocytosis below 50 cells/mm$^3$ in the CSF is highly suggestive of an acute episode of ON in the context of MS [159,160].

The role of metabolomics in the onset and progression of MS has also been studied in recent years [161]. Thus, based on the hypothesis that metabolomics highlights a series of metabolic alterations encountered in patients with severe forms of MS, it' s role in the establishment of therapeutic profiles has been studied in order to assess the degree of response to the therapy administered [162,163]. This technique allows for the analysis of a variety of small molecules below 1500 Da found in various bodily fluids, such as CSF, serum, plasma, or urine [164].

Reinke et al. [165] analyzed the CSF from 15 patients with MS and 17 from a control group and concluded that patients from the first group had energy and phospholipid metabolism alterations, which led to increased levels of choline, myoinositol, and threonate on one hand and on the other hand decreased levels of 3-hydroxybutyrate, citrate, phenylalanine, 2-hydroxyisovalerate, and mannose. In a similar study, Lutz et al. [166] demonstrated that elevated lactate and reduced phenylalanine in CSF levels contribute to the maintenance of pro-inflammatory status in MS.

### 3.4. Optical Coherence Tomography

The optic nerve is the most "visible" part for investigation in the CNS, and the fact that visual function can be measured objectively makes ON an important model for research into CNS inflammatory disease. The comparison "the eye as a window to the brain" became accurate when, for example, ON was diagnosed by optical coherence tomography (OCT) [167].

OCT is a marker of CNS axonal loss. OCT highlights a series of imaging parameters based on which correlations are made between neuronal loss and the degree of associated visual dysfunction. Several prognostic markers have been proposed, one of the most widely used being a thinning of the RNFL and the GCL ganglion cell layer that assesses the dynamic evolution of MS patients [168]. Trip et al. reported a 33% reduction in peripapillary

RNFL thickness in eyes with a history of ON and incomplete recovery. There was a 27% reduction in the affected eyes compared to the unaffected fellow eyes [169].

Similar clinical results have been reported by Frohman et al. [127] who demonstrated reduction of RNFL in patients with recurrent ON as well as in those previously diagnosed with MS. The OCT measurements showed both axonal loss and retinal ganglion cell loss and are able to predict both visual recovery or impaired visual function [132,170].

Saidha et al. [171] demonstrated that OCT facilitates the identification of pathological changes at the retinal level, the objectification of some inner and outer nuclear layer pathology associated with an advanced degree of disability and therefore with an increased severity of MS. The role of OCT in assessing axonal integrity has previously been demonstrated by Burkholder et al. [172]. Based on the pathophysiological concept that the macula contains an increased density of neuronal structures, measurement of macular thickness and volume allows indirect assessment of its properties. The same group of investigators demonstrated that pre-papillary thinning of the RNFL and inner macular volume loss are common imaging findings in MS patients with no history of ON [172]. Scanning laser polarimetry can be used as an alternative to the imaging methods presented above, with reported results showing detection sensitivity lower than OCT of lesions at 1 month (65% vs. 54%) and similar at 3 months (58% vs. 60%) [173].

OCT also facilitates the differential diagnosis between ON and myelin oligodendrocyte glycoprotein antibody associated disorder (MOGAD). Thus, while pRFNL thickening is above 5 μm in all patients with MOGAD, in MS, only 54% of cases have this associated change [174].

*3.5. Transorbital B-Mode Ultrasonography*

Transorbital B-mode ultrasonography indirectly assesses the associated inflammatory status of patients with ON, associating a narrowing of the retrobulbar portion of the optic nerve in patients with recurrent ON [175]. Despite increased sensitivity and easy accessibility, further clinical studies are needed to identify imaging parameters with prognostic value for progression to MS [175].

**4. New Therapeutic Targets**

Over the past decades, researchers in the field have been constantly concerned with identifying new molecules to explain the pathophysiological mechanisms underlying the connection between ON and MS [176]. In the era of polymedicine, the identification of effective therapeutic molecules with a reduced degree of interaction with the medication of other pathologies (especially those with cardiological target) [177–179]. CSF analysis revealed the presence of central nervous system autoimmune markers such as glial fibrillary acidic protein-IgG, with diagnostic, therapeutic, and prognostic roles alike [180]. The identification of this biomarker in the CSF suggests the presence of an autoimmune pathology, often paraneoplastic, with a high chance of a favorable therapeutic response to immunotherapy [181].

New pathological antibodies, notably against aquaporin-4 and, more recently, myelin oligodendrocyte protein, represent topics of interest to researchers in the field. Discovery of IgG1 antibodies directed against astrocyte water channel protein aquaporin 4 (AQP4) are involved in the pathophysiological mechanisms of ON in MS [182]. Identification of these autoantibodies has been more frequently associated with disease recurrence or the presence of ON [183,184].

The discovery of these molecules has had associated therapeutic value, with a number of potential new drugs being developed, such as aquaporumab (non-pathogenic antibody blocker of AQP4-IgG binding) [185,186]. Sivelestat (neutrophil elastase inhibitor) [187–189] and eculizumab (complement inhibitor) complete the list of molecules under investigation in various clinical trials at the moment [185].

Sodium channel blockade have also been proposed as potential therapeutic targets due to their role in energy metabolism in neuroinflammatory diseases [190].

Digitalization and technological advances over the last decade have enabled the discovery of new immunosuppressive agents and the development of monoclonal antibodies which, when administered, induce a superior therapeutic response and thus improve patient prognosis [176]. Mesenchymal stem cell therapy is a promising research direction, with promising clinical results in small group clinical trials [191–193]. This therapy has an anti-inflammatory effect and potentiates remyelination, but it is limited in its use in terms of identifying the anatomical site of the lesion in the optic nerve or retina [194].

## 5. Conclusions

The molecular mechanisms underlying the onset and progression of ON in patients with MS are extremely varied, incompletely elucidated to date, and continue to represent research challenges. Further clinical studies are needed to establish whether axonal degeneration is a consequence of demyelination or an independent process. Advances in technology have led to the refinement of diagnostic methods in ON and thus to increased diagnostic accuracy. Detecting the onset of axonal degeneration would be essential in establishing therapeutic behavior. Additionally, the identification of molecular mechanisms that favor remyelination would be a second direction for the therapeutic approach.

**Author Contributions:** Conceptualization, M.A.C. and D.L.Ș.; methodology, C.S., R.A.S. and C.M.B.; validation, D.L.Ș., C.S. and R.A.S.; writing—original draft preparation, M.A.C. and D.L.Ș.; writing—review and editing, C.S., R.A.S. and C.M.B. All authors contributed equally to this manuscript. All authors have read and agreed to the published version of the manuscript.

**Funding:** This research received no external funding.

**Institutional Review Board Statement:** Not applicable.

**Informed Consent Statement:** Not applicable.

**Data availability statement:** Not applicable.

**Conflicts of Interest:** The authors declare no conflict of interest.

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
