# Peer review of "Optic Neuritis in Multiple Sclerosis—A Review of Molecular Mechanisms Involved in the Degenerative Process"

_cimb, doi:10.3390/cimb44090272_

Round 1

Reviewer 1 Report (Previous Reviewer 1)

This narrative review manuscript is a rewrite to summarize the pathogenesis and molecular mechanisms of optic neuritis. This revised manuscript indeed improves the overall quality; however, this version still requires a lot of restructuring to enhance the readability. For a cohesive narrative review article, it is important to define a precise rationale to address specific questions and communicate with the readers by a well-organized structure. Unfortunately, this current version misses both key elements.

1. Abstract, Lines 19-21: This sentence sounds like a beginning sentence.

2. Introduction, Lin 71: Please indicate the period of research included in the literature search, but not the time the authors performed the search.

3. Line 104: Suggested removing "molecules or"

4. Line213: Suggested adding few sentences to define what is MS triad and make a connection to the following sections.

5. Line 231: Please expand “the validity of EAE” a little bit more.

6. The sub-headings from lines 213-346 are confusing, with and without bullets, and the connections to each other are unclear. It looks like the authors are trying to introduce some experimental models capable of capturing human MS diseases, but unfortunately the flow is difficult to follow without proper connection sentences. Suggested restructuring the contents and using numbering system to organize the sections better.

7. Line 252: Not sure of the purpose of "Finally".

8. Suggested moving Lines 284-288 to line 283 since they are related.

9. Is “Axonal and neuronal degeneration” (starting from line 333) part of MS triad (demyeolination, atrophy, remyelination)? if not, this should be a separate section. If it is, a connection is missing between this and above sections.

10. Section 3, line 413: The reviewer did not understand the meaning of the title.

11. Line 414: Does neuroimaging refer to only MRI (based on Figure 1)?

12. Figure 1: Are these three methods required for the diagnosis? Can these methods be conducted in parallel? The way presented in this figure suggests there is an order needed for diagnosis, either starting with MRI and going outward, or beginning with VEP and going inward. If so, the main text should follow the order to support the idea (MRI, CFE, FEP or reverse). If no specific order is needed, then this figure is meaningless and can be removed.

13. Figure 2 is confusing. A clear figure legend to explain the purpose and procedure would be helpful. The associated description in main text (lines 489-491) does not help with understanding Figure 2 as well. Did the authors imply that classic biomarkers are driving the diagnosis toward a completely different direction compared to proposed biomarkers?

14. Line 524: How does OCT fit in this diagnosis paradigm? It is not included in Figure 1.

15. Line 530: OCT is a technique, not a biomarker for a disease or symptom.

16. Line 558: Ultrasonography is a different tool and should not be included in OCT section.

17. Section 4: How do these new therapeutic targets facilitate "promoting remyelination" as stated in the end of abstract?

18. Lines 568-572: These sentences are actually more relevant to Figure 2.

19. Line 585: Missing reference for sodium channel blockage.

20. Section 5 can be combined with section 4 and expanded with details.

Author Response

Dear reviewer,

Thank you for your pertinent and well addressed remarks as well as for the re-review of our manuscript. We hope that the changes made correspond to the high academic level of the journal.  We have revised the manuscript in accordance with the suggestions made. Please find below a detailed point-by-point reply to the comments.

Comment 1. Abstract, Lines 19-21: This sentence sounds like a beginning sentence.

Response: We have reorganized the abstract and hope that the new paragraph will respect the recommendations made

Comment 2. Introduction, Lin 71: Please indicate the period of research included in the literature search, but not the time the authors performed the search.

Response: We revised the paragraph and mentioned that the studies included were published between 1970 and July 2022.

Comment 3. Line 104: Suggested removing "molecules or"

Response: We have revised the paragraphs mentioned.

Comment 4. Line213: Suggested adding few sentences to define what is MS triad and make a connection to the following sections.

Response: We introduced a new paragraph in which I presented the central elements of the triad in multiple sclerosis and the importance of their research.

Comment 5. Line 231: Please expand “the validity of EAE” a little bit more.

Response: We have revised the paragraphs mentioned.

Comment 6. 6. The sub-headings from lines 213-346 are confusing, with and without bullets, and the connections to each other are unclear. It looks like the authors are trying to introduce some experimental models capable of capturing human MS diseases, but unfortunately the flow is difficult to follow without proper connection sentences. Suggested restructuring the contents and using numbering system to organize the sections better.

Response: We have replaced the numbering system with a numeric one to facilitate the understanding of the content

Comment 7. Line 252: Not sure of the purpose of "Finally".

Response: We thought it appropriate to delete the word to facilitate the reading of the manuscript.

Comment 8. Suggested moving Lines 284-288 to line 283 since they are related.

Response: We have reorganised the information on NgR into one paragraph.

Comment 9. Is “Axonal and neuronal degeneration” (starting from line 333) part of MS triad (demyeolination, atrophy, remyelination)? if not, this should be a separate section. If it is, a connection is missing between this and above sections.

Response: The parahraph mentioned is part of the MS triad and was connected by it through the numbering system.

 Comment 10. Section 3, line 413: The reviewer did not understand the meaning of the title.

Response: With this title we wanted to highlight the importance of broad symptomatology and the link between clinical status and the subsidiary pathophysiological mechanisms responsible for their occurrence.

Comment 11. Line 414: Does neuroimaging refer to only MRI (based on Figure 1)?

Response: Neuroimaging involves both MRI and OCT, but in the figure we have chosen to mention only MRI based on the superior results it offers, being recognized as a primary imaging exploration in these patients.

Comment 12. Figure 1: Are these three methods required for the diagnosis? Can these methods be conducted in parallel? The way presented in this figure suggests there is an order needed for diagnosis, either starting with MRI and going outward, or beginning with VEP and going inward. If so, the main text should follow the order to support the idea (MRI, CFE, FEP or reverse). If no specific order is needed, then this figure is meaningless and can be removed

Response: We wanted the image used to suggest the associative role of the investigations proposed to establish the diagnosis, there being no necessary order to be performed.

Comment 13. Figure 2 is confusing. A clear figure legend to explain the purpose and procedure would be helpful. The associated description in main text (lines 489-491) does not help with understanding Figure 2 as well. Did the authors imply that classic biomarkers are driving the diagnosis toward a completely different direction compared to proposed biomarkers?

Response: No. Clinical studies in the literature that have treated these subjects recognize the value of classical biomarkers. Recent research has proposed a number of new molecules with a potential role in early diagnosis, but further large-scale clinical validation studies are needed. We considered it appropriate to present these as future research perspectives in the field.

Comment 14. Line 524: How does OCT fit in this diagnosis paradigm? It is not included in Figure 1.

Response: OCT provides a range of imaging information correlated with neuronal loss and the degree of associated visual dysfunction. Figure 1 highlights the main methods used for diagnosis.

Comment 15. Line 530: OCT is a technique, not a biomarker for a disease or symptom.

Response: We have corrected the paragraph mentioned.

Comment 16. Line 558: Ultrasonography is a different tool and should not be included in OCT section.

Response: We revised the paragraph mentioned.

Comment 17. Section 4: How do these new therapeutic targets facilitate "promoting remyelination" as stated in the end of abstract?

Response: We have introduced new paragraphs on the remyelination process and the therapeutic agents that can promote this process.

Comment 18. Lines 568-572: These sentences are actually more relevant to Figure 2.

Response: We revised the paragraph and we changed the position of the mention in the text of figure 2.

Comment  19. Line 585: Missing reference for sodium channel blockage.

Response: We have inserted the bibliographic reference

Comment 20. Section 5 can be combined with section 4 and expanded with details.

Response: We combined the sections as recommended.

We look forward to hearing from you.

Yours sincerely,

Dr. Delia Salaru

Reviewer 2 Report (Previous Reviewer 2)

The manuscript has been improved but the problem still persists. However, the manuscript does not make an impact on the scientific community, There are no conceptual contributions from the authors except that there is collated information from various articles.

Author Response

Dear reviewer,

Thank you for your remarks as well as for the re-review of our manuscript. We hope that the changes made correspond to the high academic level of the journal.  We have revised the manuscript in accordance with the suggestions made and we have included new paragraphs to complement the data previously presented as well as new research directions associated with the topic. 

We look forward to hearing from you.

Your sincerely,

Dr. Delia Salaru

Round 2

Reviewer 1 Report (Previous Reviewer 1)

The revised version certainly shows the improvement of the quality of the manuscript. The reviewer greatly appreciate it. However, the reviewer did not agree that Figures 1 and 2 would add values to the manuscript though; these figures are rather confusing without any changes. The reviewer, again, strongly suggested the following options: 1) completely remove these two figures; 2) add clear figure legends to demonstrate the purpose and the connection among all factors; 3) modify the figures to avoid the confusion with clear figure legends. 

Author Response

Dear reviewer,

Thank you for your suggestion. We have replaced the two figures from the original manuscript with a simple one in which we have included all the diagnostic methods described in the text.

We look forward to hearing from you.

Yours sincerely,

 Dr. Delia Salaru

This manuscript is a resubmission of an earlier submission. The following is a list of the peer review reports and author responses from that submission.

Round 1

Reviewer 1 Report

This narrative review manuscript summarizes the pathogenesis of optic neuritis. Overall, this manuscript requires a better restructuring to improve the readability. The authors failed to adequately describe the rationale for their review in the context of what is already known about the addressed questions. Many models or events mentioned in this manuscript are related to CNS-degeneration in general, but not specific to optic neuritis and/or multiple sclerosis. References listed are not up to date; only 1 reference is published in 2021 and none is from 2022.

  1. Lines 20-25 in ABSTRACT are exactly the same as Lines 44-50 in INTRODUCTION. Abstract should be a highlight and/or summary of the entire study, not a repeat of INTRODUCTION (background information). This rule shall also apply for a narrative review article.
  2. The flow is difficult to follow. Re-organizing the content is required. e.g. Section 2.2-2.8 describe various types of immune cells. However the correlation between these cells and MS pathogenesis is not defined until section 2.9. Additionally, it sounds like these cells are only involved in inflammatory phase, while CD8 T cells were often discussed in later phases.
  3. Many sentences are similar to the reference cited. Here are just two examples:
  • Lines 347-350 are very similar to the reference 75 which is a review article. Here is the original quote: "EAE-induced mice treated with pentamidine isethionate (PTM), an antiprotozoal drug shown previously to block S100B activity, had reduced EAE severity score in the preclinical, onset, and remission phases during the disease course, which correlates with the reduction of Ifng and Tnfa expression in the brain, as well as the decrease of NOS activity."
  • Lines 433-437 are similar to the discussion section in reference 87, a review article. The original sentences are: "To test the hypothesis that NMDA receptor activation during the induction phase of AON, prior to inflammatory demyelination of the optic nerve (Sühs et al., 2014), leads to these changes, the potent NMDA receptor blocker MK-801 was applied by intravitreal injection during AON. Indeed, this resulted in a stabilization of optic nerve actin dynamics and a restoration of visual functions, along with neuroprotection of RGCs. This is further supported by the observations that the retinal calcium increase which occurs during this same period during AON (as indicated by manganese-enhanced MRI; Hoffmann et al., 2013), could be significantly reduced by MK-801 treatment." The authors rephrased these sentences at the minimal level and failed to cite two original studies.
  1. The full name shall be provided for the acronym upon first mention in the text.
  2. Section 2.2: Lack of definition. e.g outside-in theory and inside-out hypotehsis both require a much more detailed explanation.
  3. The rationale of EAE is unclear in section 2. Although EAE patients may exhibit optic neuritis, their correlation should be clearly defined here. Many mechanisms and/or models mentioned in section 2.3-3.2 are EAE-related, not necessarily MS-related.
  4. Many references are wrong and not relevant to the text. Two examples: line 103, ref 21 and line 106, ref 22.
  5. Lines 130-135 are CD4+ related discussion. Not sure how it links to section 2.4 which is CD8-focused.
  6. Suggested moving section 2.9 ahead to define pathogenesis of optic neuritis.
  7. Not sure of the purpose to describe CD8 T cell in section 3 and 5 as a leading paragraph.
  8. Section 3.2. title is exactly the same title as reference 9 and the rationale of this section is unclear. Shall clearly define RR.
  9. Lines 280-342 describe current models for studying MS. However, these sections are not relevant to demyelination and axonal loss. Not sure why these were under section 3.3. Suggested making these paragraphs as a stand-alone section.
  10. Line 285: If EAE can't fully represent MS, the authors shall discuss or explain why and how the half of this manuscript is based on EAE.
  11. Line 309: Apparently, cuprizone treatment is not the "final" model the authors discussed.
  12. Line 317: The title is misleading. LPC is an endogenous lisophospholipid; however, the LPC-induced model is "exogenous exposure".
  13. Lines 334-342 are not relevant to LPC model.

Reviewer 2 Report

The review article is on an interesting topic. However, the manuscript is poorly written, and the literature has been poorly arranged and discussed. There are no interpretations and conclusions by the authors and mostly the conclusions by the authors of different papers are compiled. Often there is a disconnect between sections. Transitions from one section to the other are also not linked properly. the Neuronal part of optic neuritis does not start until several pages from the introduction. Several paragraphs and just 1-2 sentences as if it is taken from just one paper., then another paper in the next paragraph. A proper discussion on reports from various groups and their combined conclusions is missing. Overall, as written, this is not at all useful for readers. A few specific comments are below:

  1. MS is a neurological problem affecting the CNS and characterized by inflammatory, demyelinating and neurodegenerative changes. While inflammation is well studied in MS-associated pathologies, there is increasing evidence demonstrating that neurodegeneration is critically involved. However, the review has not discussed any aspects of neurodegeneration in optic neuritis in the introduction.
  2. Authors have stated multiple times about step-by-step mechanisms/events in MS-induced ON. However, the exact sequence of molecular events is still not completely studied. Authors need to generalize the statement.
  3. The manuscript is lacking proper structure/organization. Sections are lacking connections in their contents.
  4. Pathophysiology of optic neuritis should be a separate section
  5. No diagrams included
  6. Many abbreviations are not expanded when used for the first time
  7. “Mature OLGs that have not been demyelinated are unable to form new myelin sheaths” is an incorrect statement. OLGs are myelin-forming cells and do not demyelinate.
  8. Models of ON or MS should be discussed as a separate section.
  9. Overall, the review is poorly written and organized.